# EGGen: Image Generation with Multi-entity Prior Learning through Entity Guidance

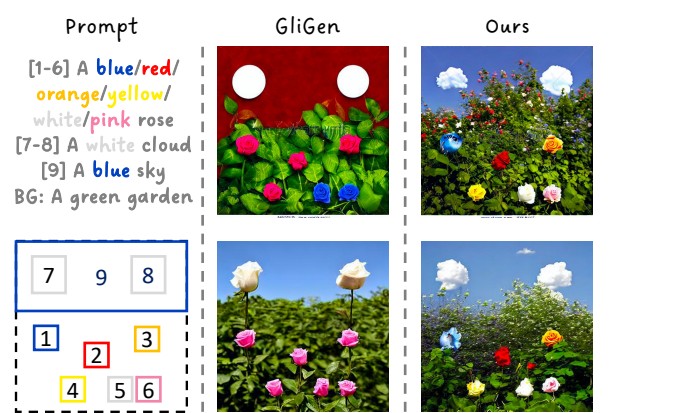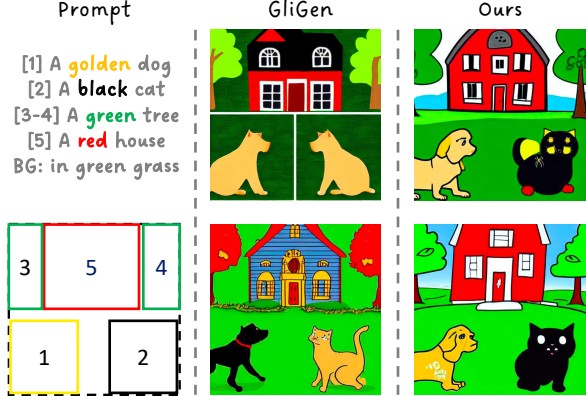

(a) Multiple entities with multiple attributes in a realistic style.

(b) Multiple entities with multiple attributes in a cartoon style.

Figure 1: EGGen's generations using entity-level text prompts and predicted layout for image generation. Numbers with brief prompts and boxes are displayed on the left image, which refer to specific entities. The layout of boxes can either be predicted by Large Language Models (LLMs) or manually input.

## ABSTRACT

Diffusion models have shown remarkable prowess in text-to-image synthesis and editing, yet they often stumble when tasked with interpreting complex prompts that describe multiple entities with specific attributes and interrelations. The generated images often contain inconsistent multi-entity representation (IMR), reflected as inaccurate presentations of the multiple entities and their attributes. Although providing spatial layout guidance improves the multi-entity generation quality in existing works, it is still challenging to handle the leakage attributes and avoid unnatural characteristics. To address the IMR challenge, we first conduct in-depth analyses of the diffusion process and attention operation, revealing that the IMR challenges largely stem from the process of cross-attention mechanisms. According to the analyses, we introduce the entity guidance generation mechanism, which maintains the integrity of the original diffusion model parameters by integrating plug-in networks. Our work advances the stable diffusion model by segmenting comprehensive prompts into distinct entity-specific prompts with bounding boxes, enabling a transition from multi-entity to single-entity generation in cross-attention layers. More importantly, we

introduce entity-centric cross-attention layers that focus on individual entities to preserve their uniqueness and accuracy, alongside global entity alignment layers that refine cross-attention maps using multi-entity priors for precise positioning and attribute accuracy. Additionally, a linear attenuation module is integrated to progressively reduce the influence of these layers during inference, preventing oversaturation and preserving generation fidelity. Our comprehensive experiments demonstrate that this entity guidance generation enhances existing text-to-image models in generating detailed, multi-entity images.

## CCS CONCEPTS

• **Computing methodologies** → **Computer vision**.

## KEYWORDS

Diffusion model, Text-to-image Generation, Multi-entity Generation

**ACM Reference Format:**
Anonymous authors. 2024. EGGen: Image Generation with Multi-entity Prior Learning through Entity Guidance. In *In Proceedings of 32th ACM International Conference on Multimedia (MM '24), 28 October - 1 November 2024, Melbourne, Australia.* ACM, New York, NY, USA, 9 pages. https://doi.org/XXXXXXX.XXXXXXX

## 1 INTRODUCTION

The domain of text-to-image synthesis has experienced significant progress, particularly through the integration of diffusion models [2, 3, 8, 18, 22, 23, 29]. These models have demonstrated exceptional proficiency in creating images that are both highly realistic

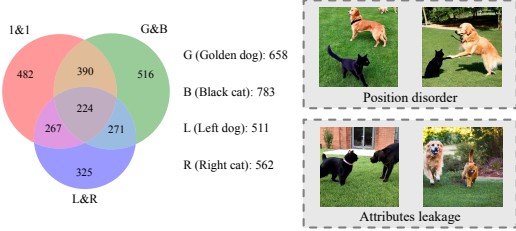

(a) Position disorder and attribute leakage.

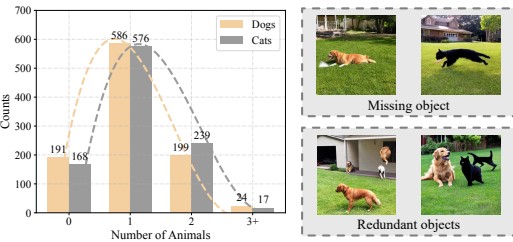

(b) Occurrences of inaccurate entities.

**Figure 2: The statistics of 1000 examples generated by SD V1.5 with the prompt of "A golden dog on the left and a black cat on the right are playing in the yard".**

and varied, based on text prompts. Nevertheless, despite their outstanding performance, diffusion models like Stable Diffusion [22] sometimes face challenges in accurately interpreting prompts when they involve complex arrangements of *multiple entities*. These issues mainly manifest as entity position disorder, attribute leakage (i.e., misallocated attributes), and inaccurate presentation of entities (e.g., missing or redundant entities), as illustrated in Figure. 2. These issues collectively lead to a mismatch between the intended multi-entity compositions of the prompts and the actual image outputs, highlighting the challenge of **I**nconsistent **M**ulti-entity **R**epresentation (**IMR**).

Previous methods address the IMR issue mainly by relying on pre-defined bounding boxes (i.e., layout) to constrain the multi-entities' position and number at image spatial domain [1, 12, 13, 15, 26, 30]. While these methods afford SD models the capability to take care of the entities' specific positions and achieve improvement in the results, it is still challenging to achieve natural entity placement and precious attribute allocation and presentation without leakage. Although the bounding box-based hard assignment gives a direct restriction on the coordinates of the entity in the spatial domain, the interactions (e.g., the cross-attention operations) of the long/complex text prompts and visual representations are still handled as a whole, leading to mixture and confusion in the results [12], as discussed in Figure 3. Directly applying bounding box-based constraints on the image spatial domain may also result in unnatural artifacts on the generated images [13, 30].

The objective of our research is the generation of multiple entities from complex text descriptions with bounding boxes, enhancing precision by addressing IMR issues in generative models through fine-tuning adaptors while keeping the parameters of the original diffusion model fixed. To understand potential IMR issues during generation, we first analyze the cross-attention operations of text-based generative models, conducting detailed analyses of token-wise and step-wise cross-attention maps (see Sec. 3). The token-wise analysis indicates that the prompt tokens for different entities and their specific attributes are aggregated together to control the generation of visual representations in SD, which can easily lead to an **entity coupling** phenomenon marked by mismatched entity types and attributes blending across entities when the prompts are complex. Our step-wise analysis of the diffusion process reveals the problem of **entity prematurity** – the visual patterns, e.g., the positioning and characteristics of entities, are

usually established prematurely in the early steps of the diffusion process, with low-resolution attention maps. The improperly mixed tokens of different entities (because of the *entity coupling* issue) can lead to improper attention maps at an early stage (e.g., cross attention at first several steps), resulting in generated images with IMR, due to the *prematurity*.

Building on the outlined observations, we introduce an **E**ntity **G**uidance **Gen**eration (**EGGen**) mechanism to address IMR issues within cross-attention layers. To handle the complex prompts including descriptions of multiple entities, we first strategically segment a comprehensive prompt into distinct entity-specific prompts with bounding boxes by the LLM, facilitating a shift from multi-entity to single-entity generation within cross-attention layers. To counteract entity coupling, we introduce **E**ntity-centric **C**ross-**A**ttention (**ECA**) layers focused on individual entity prompts instead of the general cross-attention operation, thereby safeguarding each entity's uniqueness and correctness of the type. Simultaneously, **G**lobal **E**ntity **A**lignment (**GEA**) layers serve as the refinement of cross-attention maps within the standard cross-attention layers to use multi-entity priors (Holistically-Nested Edge Detection (HED) [27]) as a ground truth for guiding accurate entity positioning and attribute delineation. Targeting the entity prematurity, a **L**inear **A**ttenuation (**LA**) module is integrated to linearly decrease the impact of ECA and GEA layers as the step increases when inference, preventing oversaturation and ensuring generation fidelity. In our experiments, our EGGen model demonstrates precise positional control and attribute accuracy in generating multiple entities through entity guidance, as evidenced on T2I-CompBench [9] and visual case studies. Especially, Figure 1 illustrates the generation of multiple entities with precise attribute control in both realistic and cartoon styles.

The key contributions of our work are summarized as follows:

- We explored the underlying causes of IMR issues through the study of token-wise and step-wise attention maps, identifying the effect of entity coupling and entity prematurity.
- Our EGGen model advances a stable diffusion approach by segmenting comprehensive prompts into entity prompts with bounding boxes, transferring the multi-entity generation to single-entity generation within cross-attention layers.
- Combined with ECA, GEA, and LA, our EGGen model achieves precise positional control and attribute accuracy in the generation of multiple entities.

## 2 RELATED WORK

**Text-to-Image Generation**. In the swiftly evolving field of text-based image generation, an array of model architectures and learning paradigms have surfaced, as evidenced by a series of pivotal studies [2–5, 10, 17, 19–21, 28, 29, 32]. Initially, GAN-based models [19, 28, 32] were at the forefront, setting foundational benchmarks for the quality and diversity of the images. Recently, the advent of diffusion models [18, 22, 23] marked a significant leap forward, enhancing the fidelity and realism achievable in text-to-image generation. These models operate on the principle of structured denoising [8] with latent diffusion [22], which begins with initializing random noise in a latent space. This noise is then systematically refined through a denoising process, transforming it into visually detailed images by incorporating textual conditions. This method enhances both diversity and realism in generated images, making latent diffusion models a powerful player in generative AI.

**Multi-entity Generation.** Multi-entity synthesis is an area of significant interest due to its potential and broad applications in industries. Most efforts [1, 12, 13, 15, 26, 30] to address the challenges of diffusion models in accurately representing multiple entities with special attributes. For instance, GLIGEN [12] adopted bounding box coordinates as grounding tokens and integrated them into a gated self-attention mechanism to enhance positioning accuracy. Furthermore, the LLM-grounded diffusion model [13] used DDIM inversion to create initial latents for each entity and then applied the GLIGEN model for precise layout arrangement. Detect guidance [15] integrated a latent object detection model to separate different objects during the generation process, then masked the conflicting prompts and enhanced related ones. Despite existing methods of generating images with correct positions, challenges persist, especially in generating images that accurately blend attributes from multiple entities. Our work is focused on investigating the underlying reasons behind the challenges of synthesizing multiple entities and conducting a divide-and-conquer mechanism to enhance entity-centric modeling in cross-attention operations.

## 3 ANALYSES ON IMR CHALLENGE

**Inconsistent Multi-entity Representation.** Our approach begins with a thorough examination of the creation of multiple entities during the diffusion process. To understand the issues of the multi-entity generation, we conduct a statistical analysis on 1000 generated images using the prompt "*A golden dog on the left and a black cat on the right are playing in the yard.*" with the SD V1.5 model, selecting different random seeds for each trial. The diverse outcomes of this experiment are illustrated in Figure 2, which reveals approximately 50% cases with **inaccurate entities**, 50% cases with **attributes leakage**, and around 70% cases with **position disorder**. These findings indicate a challenge of inconsistent multi-entity representation, often resulting in a low likelihood of fully adhering to the intended multi-entity compositions of the prompts.

**Entity Coupling.** Building on the insights from Hertz et al. [6] regarding the $32 \times 32$ resolution of cross-attention maps, we further investigate certain phenomena in diffusion models. This exploration involves analyzing token-wise attention maps within the U-Net architecture, as demonstrated in Figure 3, aiming to uncover the token

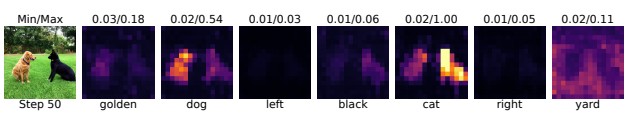

(a) A good case: each entity and its attributes have stronger contours.

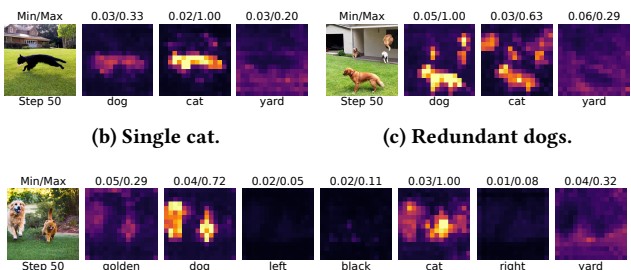

(b) Single cat.     (c) Redundant dogs.

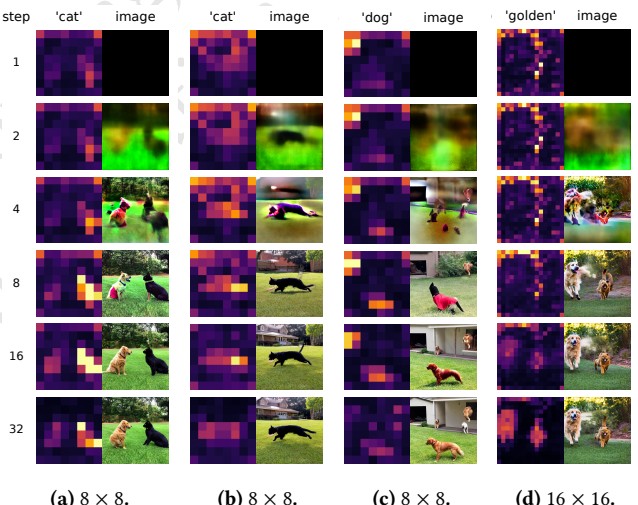

(d) Mismatch: one entity has a strong contour of the other entity.

**Figure 3: Token-wise attention maps ($32 \times 32$) across all timestamps of a diffusion process, showcasing semantic relationships that exhibit the entity coupling of tokens.**

**Figure 4: Step-wise attention maps in the low-resolution layers by inference steps ($\{1, 2, 4, 8, 16, 32\}$), showcasing the entity prematurity of cross-attention.**

(a) $8 \times 8$.  (b) $8 \times 8$.  (c) $8 \times 8$.  (d) $16 \times 16$.

impact of inconsistent representations of multiple entities. Our observations highlight semantic relationships that exhibit the **entity coupling of cross-attention** across tokens and their impact on the accuracy of generating images with multiple entities. In good case (Figure 3a), when one entity exhibits relatively weak signals in the attention map of another, the coupling between entities is deemed acceptable and not harmful, as each entity and its attributes are delineated by its stronger signal in the attention maps. In contrast, failing cases reveal: (1) **Position disorder**: The attention map's marked insufficiency in responding to spatial tokens like *left* and *right* underscores diffusion models' difficulties with spatial interpretation (Figure 3a and 3d). (2) **Inaccurate entities**: The appearance of unusual targets in the scene complicates the model's capacity to

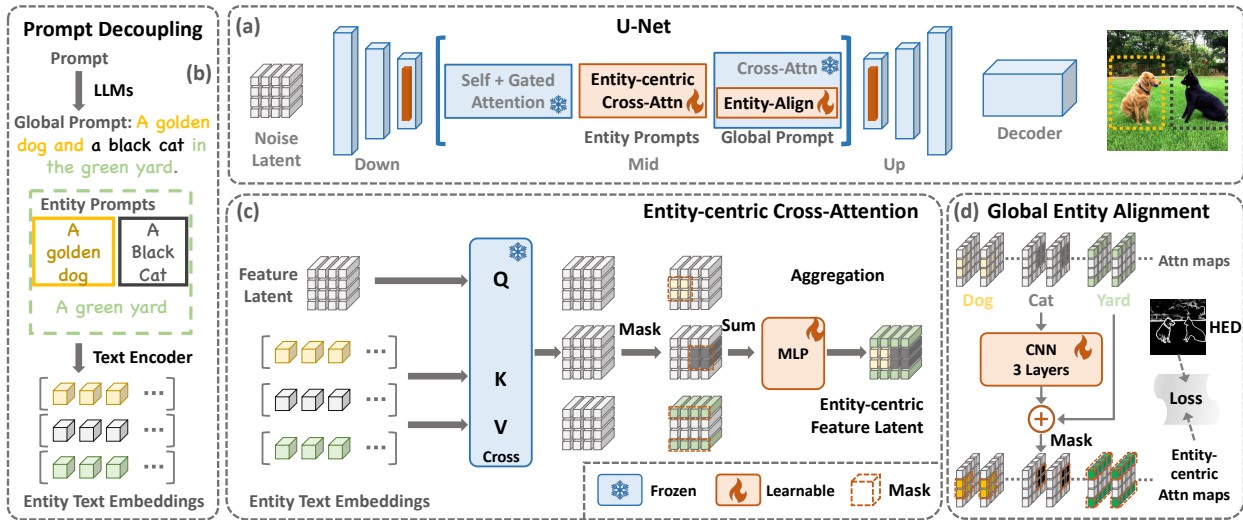

Figure 5: Overview of the proposed learnable entity guidance generation in the frozen pre-trained latent diffusion model. (a) The up-middle part indicates the proposed ECA and GEA plugged into the U-Net framework of GLIGEN model; (b) The left part shows the process of prompt decoupling; (c) The down-middle part indicates the divide and conquer process of the ECA layer based on the entity prompts; (d) The right part shows the GEA layer refines cross-attention maps with the global prompt.

distinguish between entities, while entity coupling causes loss of control over the presence or absence of targets (Figure 3b and 3c). (3) **Attribute leakage**: The entity coupling incorrectly aligns the entity of the "cat" strongly on the map of the "dog" token, leading to the *golden* attribute mistakenly associating with cat (Figure 3d).

**Entity Prematurity.** Previous research [1, 25] has highlighted that the minimum resolution of cross-attention layers dictates contour definition, whereas higher resolution layers are responsible for details. Our investigation extends these findings by examining the step-wise attention map in the low-resolution layers, as shown in Figure 4. From Figure 4, we observe that positioning and quantity of entities are established at a low resolution early in the process, sometimes as early as step 1, while detailed attributes like colors then play a pivotal role in later stages to refine the image's appearance. Furthermore, the phenomenon of entity coupling also persists throughout the inference process, contributing to inaccurate entities. The entire process appears to resemble the **Entity Prematurity of cross-attention**.

Based on both token-wise and steps-wise analysis on attention maps, we learn that the challenge of inconsistent multi-entity representation primarily arises from the entity coupling and prematurity of cross-attention, resulting in the inaccuracy of multiple entity generation. In response, we propose the Entity Guidance Generation (EGGen) strategy to tackle these specific challenges.

## 4 PROPOSED APPROACH

In this section, we present an overview outlining the comprehensive mechanism of our approach. This is followed by a detailed examination of the entity-centric cross-attention and the alignment of attention refinement. We conclude with an explanation of the overall optimization strategy.

### 4.1 Overview

In the task of text-to-image generation, diffusion models aim to accurately transform textual prompts into corresponding images. The latent diffusion architecture integrates textual information $\mathbf{y}$ into the image synthesis process via a cross-attention layer. Initially, textual prompts are encoded into embeddings $\mathbf{s} \in \mathbb{R}^{n \times d}$, which are then mapped through the cross-attention mechanism, involving query $\mathbf{Q}_i \in \mathbb{R}^{hw_i \times d_i}$, key $\mathbf{K}_i \in \mathbb{R}^{n \times d_i}$, and value $\mathbf{V}_i \in \mathbb{R}^{n \times d_i}$ vectors, to produce attention maps $\mathbf{A}_i \in \mathbb{R}^{hw_i \times n}$. Both key and value vectors are generated from the text-conditioned embeddings. To address the IMR challenge of the SD models highlighted in Section 3, we introduce the EGGen methodology, which builds on the strengths of the pre-trained GLIGEN SD model [12], as detailed in Figure 5.

The process of the proposed EGGen can be divided into (1) **Prompt decoupling**: an LLM is utilized to reorganize the provided prompt into a global prompt, and separate entity prompts with spatial locations (marked by the bounding boxes). This organization enables the direct association of attributes with their respective entities, enhancing the model's ability to recognize each entity. These spatial locations of entities are then also fed into the gated attention to secure the precise positioning of the coordinates. (2) **Entity-centric cross-attention**: the entity-centric cross-attention layer is introduced that focuses on the entity prompts related to each entity, ensuring that the distinctiveness of each entity is maintained. Additionally, we apply box masking within each feature map to isolate sections corresponding to other entities, followed by an aggregation process yielding an integrated latent feature centered around each entity. (3) **Global entity alignment**: the global entity alignment layer is implemented alongside the original cross-attention layers that process the global prompt. The GEA serves as a refinement step, using multi-entity prior information (such as

HED images) as ground truth to guide the correct positioning of each entity and separate attributes from other entities.

In the subsequent section, we will provide a detailed exposition of these modules and their underlying rationales, alongside the overall optimization process.

### 4.2 Prompt Decoupling

Frequently, the challenge for diffusion models in accurately recognizing and attributing unique characteristics arises from prompt ambiguity, where entities and their attributes are intertwined. However, LLMs possess the capability to discern individual entities and predict the overall spatial layout of an image. Capitalizing on this strength, we decouple the prompt to reorganize the original prompt into a global prompt and entity prompts for each entity. Such segmentation facilitates a direct linkage of attributes to their corresponding entities, thereby improving the model's proficiency in distinctly recognizing and interpreting each entity.

Specifically, we employ advanced language comprehension and inferential capabilities of the LLM (such as GPT-4) to discern the entities and their attributes within the given prompt $\mathbf{y}$, leading to the generation of reorganized global prompt $\widehat{\mathbf{y}}$ and entity prompts $\widetilde{\mathbf{y}}$, expressed as:

$$\widehat{\mathbf{y}} = LLM(\mathbf{y}) = \widehat{y}^1 + \widehat{y}^2 + \ldots + \widehat{y}^N, \tag{1}$$

$$\widetilde{\mathbf{y}} = \{\widetilde{y}^j\}_{j=1}^N = \{\widetilde{y}^1, \widetilde{y}^2, ..., \widetilde{y}^N\} = \mathcal{F}_{re}(\{\widehat{y}^1, \widehat{y}^2, ..., \widehat{y}^N\}), \tag{2}$$

where $N$ signifies the total comprising $N-1$ foreground entities and one background element, and the $\mathcal{F}_{re}$ is the re-caption operation, which enables the generation of denser, fine-grained details for each entity prompt. This global prompt offers a concise summary, highlighting key attributes for each entity. The re-caption operation enables the generated prompt of denser, fine-grained details for each entity. Additionally, we enhance each entity with bounding box attributes, predicted by the language model after it assesses the image's overall layout, allowing text-to-image model to interpret prompts with greater accuracy, as detailed below:

$$\mathbf{B} = \{B^j\}_{j=1}^{N-1} = \{[start_x^j, start_y^j, end_x^j, end_y^j]\}_{j=1}^{N-1}. \tag{3}$$

An example of the prompt decoupling is illustrated on the left part of Figure 5. For further details on employing LLMs refer to the **Appendix**.

### 4.3 Entity-centric Cross Attention

Even with prompts clearly outlining entities and their attributes, diffusion models can struggle with entity coupling in cross-attention layers without a mechanism to handle this hierarchical information. In our approach, we introduce entity-centric cross-attention layers inserted ahead of the original cross-attention layers. The ECA layer shares weights with the original cross-attention, allowing for interaction between the feature latent and specific entity prompts. This ensures that each unique entity and its attributes are preserved. The process can be formulated as

$$\widetilde{\mathbf{f}}_i^E = \phi_i\left(\text{softmax}\left(\frac{\widetilde{\mathbf{Q}}_i\widetilde{\mathbf{K}}_i^T}{\sqrt{d_i}}\right)\widetilde{\mathbf{V}}_i\right), \tag{4}$$

where $i$ represents the $i$th layer cross-attention in the UNet and $\phi_i(\cdot)$ is the original Multilayer Perceptron (MLP) layer. The new queries

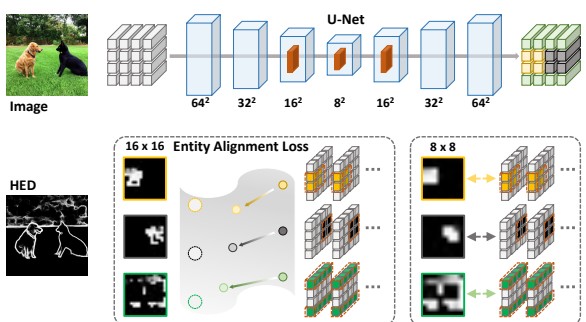

**Figure 6: Entity guidance of downscale HED soft edge with corresponding scale attention maps ($hw_1 = [16^2, 8^2]$) in each cross-attention layer of the U-Net architecture.**

$\widetilde{\mathbf{Q}}_i \in \mathbb{R}^{N \times hw_i \times d_i}$ are generated to correspond with the transformed latent representations $\mathbf{f}_i \in \mathbb{R}^{hw_i \times d_i}$, reflected by $N$ duplicates. Keys $\widetilde{\mathbf{K}}_i \in \mathbb{R}^{N \times n \times d_i}$ and values $\widetilde{\mathbf{V}}_i \in \mathbb{R}^{N \times n \times d_i}$ are created through linear projections of entity text-conditioned embeddings $\widetilde{\mathbf{s}} \in \mathbb{R}^{N \times n \times d}$, which originate from the entity prompts $\widetilde{\mathbf{y}}$.

Moreover, we utilize bounding boxes $\mathbf{B}$ as masks within the cross-attention layers to ensure accurate spatial representation of entities. These bounding boxes are resized to match the dimensions of the attention maps, effectively transforming them to a compatible size of $hw_i$, and creating a mask $\mathbf{M}_i \in \mathbb{R}^{N \times hw_i}$. We then consolidate the feature latents across the $N$ dimension after masking, and apply $\phi_i^E$ network to average the entity-centric feature latents $\widetilde{\mathbf{f}} \in \mathbb{R}^{hw_i \times d_i}$ by summarizing the input $\mathbf{f}_i$. The aggregation process is mathematically represented as:

$$\widetilde{\mathbf{f}}_i = \gamma \times \tanh(\alpha_i) \times \phi_i^E(\text{Sum}(\mathbf{M}_i \odot \widetilde{\mathbf{f}}_i^E)) + \mathbf{f}_i, \tag{5}$$

where $\gamma$ is a fixed scalar setting to 1 in training and $\alpha_i$ is a learnable scalar which is initialized as 0. This whole design of the ECA layers tackles entity coupling by isolating each entity with its designated prompt. The aggregation with bounding masks further guarantees the precision and uniqueness of each entity's depiction, emphasizing their distinct attributes.

### 4.4 Global Entity Alignment

Generating entities independently and merging them directly, without accounting for their interactions, can lead to sub-optimal integration. Merely using a cross-attention layer followed by a global prompt to interact with all entities within the same context may result in inaccuracies and attribute leakage since it does not address the problem of entity coupling. We address the sub-optimal integration by implementing a global entity alignment, as illustrated in the right part of Figure 5. This involves refining the cross-attention maps $\mathbf{A}_i \in \mathbb{R}^{m \times hw_i \times d_i}$ ($m$ represents the total number of valid tokens for N entities in the global prompt.) to better correspond with entity tokens through a CNN network $\phi_i^G$. Additionally, we replicate bounding box masks $\mathbf{M}_i \in \mathbb{R}^{N \times hw_i}$ to $\widehat{\mathbf{M}}_i \in \mathbb{R}^{m \times hw_i}$ ensuring that the attention maps for $m$ entity tokens are constrained to their specific spatial positions, thereby mitigating attribute leakage,

$$\widehat{\mathbf{A}}_i = \gamma \times \tanh(\beta_i) \times (\widehat{\mathbf{M}} \odot \phi_i^G(\mathbf{A}_i)) + \mathbf{A}_i, \tag{6}$$

where $\gamma$ is set as a fixed scalar 1 in training and $\alpha_i$ is a learnable scalar which is initialized as 0.

To further refine the attention maps for accurately capturing the intricate details of multiple entities, we employ a multi-entity prior learning strategy as guidance. We adopt HED [27] soft edge images as the prior information, which detect the contour $\mathbf{C} \in \mathbb{R}^{h \times w \times 1}$ over original images. According to the entity prematurity in Section 3, these HED images inform entity-focused attention maps $\widehat{\mathbf{A}}_i$ in cross-attention layers at $8 \times 8$ and $16 \times 16$ resolutions for processing efficiency. The choice of HED images is due to their ease of processing and richness in information, which benefits our fine-tuned layers' learning process, ensuring that the attention maps are precisely aligned. Further details on this process are depicted in Figure 6 and this alignment can be defined as:

$$\mathcal{L}^i_{hed}(\widehat{\mathbf{C}}_i, \widehat{\mathbf{A}}_i) = \frac{1}{m} \left[ 1 - \mathcal{D}(\widehat{\mathbf{C}}_i, \ \widehat{\mathbf{A}}_i) \right] , \qquad (7)$$

where $\widehat{\mathbf{C}}_i \in \mathbb{R}^{m \times hw_i \times 1}$ is a segmented contour representation by $\mathbf{B}$ from $\mathbf{C}$, resized to the target resolution of $8 \times 8$ or $16 \times 16$ and replicated to align with the $m$ dimension of $\widehat{\mathbf{A}}_i$, and $\mathcal{D}(\cdot, \cdot)$ denotes the cosine similarity function, evaluated for each token's corresponding segment at the resolution $hw_i$. By minimizing this cosine distance, we refine the focus of the entity-centric attention map on prior entity-specific information, directing the synthesis of detailed entity structures while limiting refinement to areas associated with grouped tokens.

## 4.5 Overall Optimization

Meanwhile, the denoising loss is also incorporated into the training process to further ensure the quality of the synthesized images. Therefore, the overall optimization objective can be expressed as follows:

$$\mathcal{L}_{ldm} = \mathbb{E}_{x, \epsilon \sim \mathcal{N}(0,1)}[\|\epsilon - \epsilon_\theta(z_t, t)\|^2_2] , \qquad (8)$$

$$\mathcal{L} = \lambda \sum_{i \in L} \mathcal{L}^i_{hed} + \mathcal{L}_{ldm} , \qquad (9)$$

where $L$ denotes the number of U-Net layers at resolutions of $8 \times 8$ and $16 \times 16$, and $\lambda$ denotes the entity guidance loss weight. Notably, the HED images are exclusively employed during the training phase to enhance the model's awareness of contours and are not utilized during inference.

When inference, equally weighting ECA and GEA can excessively influence the rendering of details, occasionally resulting in oversaturated images compared to traditional text-to-image models. According to entity prematurity, entities and their contours are identified early in the inference process. To mitigate this, we implement **linear attenuation** during inference, gradually reducing the $\gamma$ to 0:

$$\gamma(t) = (T_s - t)/T_s, \qquad (10)$$

where $T_s$ is the total steps of inference, commonly setting to 50 in the standard inference of the diffusion model and $t \in [1, 50]$. This strategy could well remain the entity guidance within the models while decreasing the over-saturation of the generations. After the overall optimization, the EGGen effectively addresses the issue of inconsistent multi-entity depictions, ensuring the generated images are both semantically consistent and visually detailed.

## 5 EXPERIMENTS

### 5.1 Implementation Details

**Baselines.** We utilize the layout advancements from *GLIGEN* [12] as a base model to fine-tune our approach. We further extend our comparison with a spectrum of alternative approaches. Within the realm of training-free methods, we compare: (1) *BoxDiff* [26]; (2) *Backward Guidance* [1]; (3) *LLM-grounded Diffusion* [13]. In the domain of training-based methods, our analysis encompasses: (1) *ReCo* [30]; (2) *GLIGEN* [12]; (3) *Detect Guidance* [15].

**Datasets.** We use the 414K text-image pairs as training datasets, which are reorganized by *ReCo* [30] from COCO 2014 [14] train set. To comprehensively illustrate the effectiveness of our proposed method, we adopt T2I-CompBench [9] as the test dataset, which consists of 6,000 compositional text prompts from 3 categories (attribute binding, entity relationships, and complex compositions) and 6 sub-categories (color binding, shape binding, texture binding, spatial relationships, non-spatial relationships, and complex compositions).

**Evaluation Metrics.** We follow the evaluation of T2I-CompBench [9] over the consistency between images and multi-entity prompts regarding Attribute Binding, entity Relationship, and Complex, which comprehensively utilizes various metrics, including B-VQA [11], UniDet [31] and Clip-score [7].

**Implementation Details.** The total trainable parameters are the three-layer MLP from the proposed ECA layers and the four-layer CNN network in GEA. At the same time, ECA and GEA modules are exclusively implemented at resolutions of $8 \times 8$ and $16 \times 16$. The utilized LLM is the GPT-4-Vision for its robustness and exceptional performance. During training, we use the AdamW optimizer [16] with a fixed learning rate of 0.00001 and weight decay of 0.01 for 10 epochs, and we set $\lambda = 10$ for loss control. In the inference stage, we adopt DDIM sampler [24] with 50 steps and set the guidance scale to 7.5. All experiments are performed on $8 \times$ Nvidia Tesla V100 GPUs. See the **Appendix** for more implementation details.

### 5.2 Main Results

**Qualitative Evaluation.** Given the varied capabilities of different models, we employ prompts that describe two entities to guarantee a fair comparison, and the visual comparisons are showcased in Figure 7. In the cases examined, while most approaches demonstrate the capacity to accurately position entities at appropriate coordinates, they occasionally place the wrong type of entities and assign undetermined attributes. Conversely, our method successfully positions the correct entity along with its attributes in these scenarios. For instance, the golden dog and the black cat exhibit different vivid attitudes while strictly following the prompt requirement of the dog on the left and the cat on the right. For an illustration of our EGGen's capability to generate multiple entities, please see the examples featured in Figure 1. Further visual examples can be found in the **Appendix**.

**Quantitative Evaluation.** We conduct comparisons with prior state-of-the-art (SOTA) multi-entity text-to-image models across three key compositional scenarios on the T2I-CompBench. From the results, as shown in Table 1, our approach demonstrates better

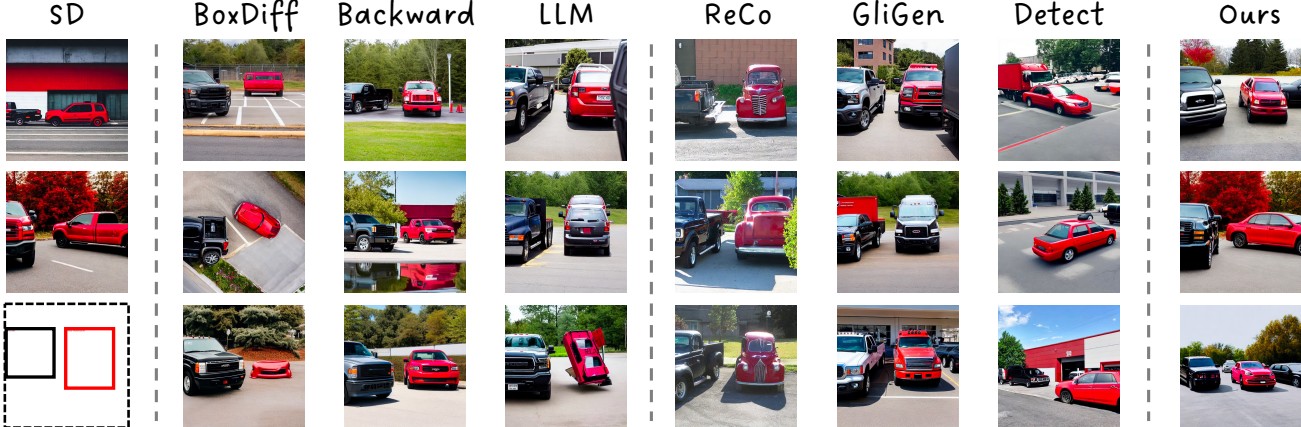

**Figure 7: Qualitative comparison with baseline methods. More examples across domains are included in the Appendix.**

Table 1: Evaluation results on T2I-CompBench. Our method demonstrates better comprehensive performance compared with other multi-entity SD-based methods.

| Model | Attribute Leakage | | | Entity Relationship | | Complex↑ |
|---|---|---|---|---|---|---|
| | Color ↑ | Shape↑ | Texture↑ | Spatial↑ | Non-Spatial↑ | |
| *SD-v1.5* [22] | 0.2365 | 0.4054 | 0.3954 | 0.1303 | 0.2864 | 0.2959 |
| *BoxDiff* [26] | 0.4153 | 0.4563 | 0.4959 | 0.2182 | 0.2621 | 0.2906 |
| *Backward Guidance* [1] | 0.3505 | 0.4085 | 0.3738 | 0.1706 | 0.2739 | 0.2619 |
| *LLM-grounded* [13] | 0.3007 | 0.5082 | 0.5071 | 0.3977 | 0.2740 | 0.2805 |
| *GLIGEN* [12] | 0.2552 | 0.4511 | 0.5097 | 0.3269 | 0.2880 | 0.2756 |
| *ReCo* [30] | 0.4059 | 0.4817 | 0.5545 | 0.2689 | 0.2856 | 0.2984 |
| *Detect Guidance* [15] | 0.4210 | __0.5122__ | 0.6136 | 0.1268 | 0.2813 | 0.3450 |
| *Ours* | __0.4586__ | 0.4946 | __0.6164__ | __0.4018__ | __0.3176__ | __0.3794__ |

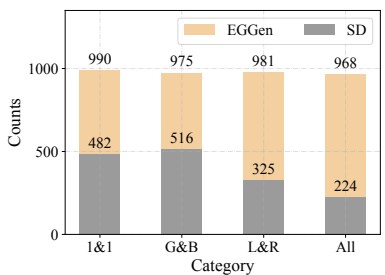

Figure 8: The comparison between the count of the different categories of 1000 examples generated by our model and SD V1.5 following the setting in Figure 2.

comprehensive performance in five scenarios, exhibiting superior fidelity and precision in aligning with the text prompts. Notably, our method achieved the highest scores in scenarios involving color and texture, underscoring its remarkable capability to precisely interpret and replicate the colors and textures described in the text inputs. These benefits from our strategy of separating the prompts into a global prompt and individual entity prompts. We utilize the ECA to process each entity singularly, preserving its uniqueness and ensuring the correct types, while the GEA refines the attention map, precisely guiding the delineation of attributes. Notably, our method slightly underperformans in the shape domain over *LLM-grounded* [13] and *Detect Guidance* [15]. The shape attribute can be inherently more complex to interpret from the text than colors or textures, so it can easily be compounded by the interaction of the mask within the GEA, causing deformation of shape. We will address this complexity in future iterations.

## 5.3 Ablation Study and Analysis

In this validation, we conducted an ablation study by individually removing various modules to assess their impact, with the results presented in Figure 9 and Table 2. The data reveals that 1) the absence of the ECA results in the sole use of the global cross-attention layer, which proves insufficient for accurately identifying and correctly positioning each entity; 2) removing GEA leads to a reliance on undifferentiated global prompts within the original cross-attention mechanism, which in turn results in attribute leakage and the generation of inaccurate entities; 3) eliminating the LA module precipitates a marked increase in image over-saturation and a discernible degradation in visual quality. This effect aligns with the observation of entity prematurity. Our ablation study highlights the critical roles of the ECA, GEA, and LA modules in enhancing the accuracy, attribute fidelity, and visual quality of our model. These findings underscore the importance of these modules in achieving superior performance in multi-entity generation.

To conclude this section, we replicate the statistical analysis experiment similar to that depicted in Figure 2, generating 1,000 examples based on the same prompt. The outcomes of this experiment are illustrated in Figure 8. In scenarios involving the generation of images with two simple entities, such as a dog and a cat, our method

Table 2: Ablation study with results on T2I-CompBench in the metrics of Color, Spatial, and Complex. *Baseline* represents the GLIGEN model; Other models represents GLIGEN model plus corresponding modules.

| Model | Color ↑ | Spatial ↑ | Complex ↑ |
|---|---|---|---|
| *Baseline* | 0.2552 | 0.3269 | 0.2756 |
| *GEA+LA* | 0.3549 | 0.3564 | 0.3142 |
| *ECA+LA* | 0.4123 | 0.3789 | 0.3325 |
| *ECA+GEA* | 0.4427 | 0.3922 | 0.3665 |
| *ECA+GEA+LA* | 0.4586 | 0.4018 | 0.3794 |

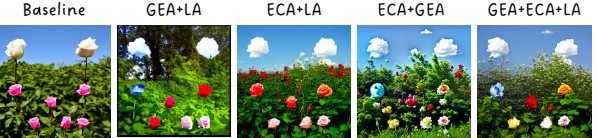

Figure 9: Visual comparison of different proposed modules.

demonstrates a high level of consistency between the text prompts and the resulting images, underscoring its robustness in addressing the challenge of inconsistent multi-entity representation.

## 6 CONCLUSION

In this paper, our study addressed the challenge of IMR in diffusion-based text-to-image synthesis after analyzing the effect of entity coupling and entity prematurity. By integrating the EGGen mechanism within cross-attention operations, we effectively improved entity positioning and attribute accuracy while maintaining generation fidelity. Our approach leverages ECA layers and GEA layers to ensure precise entity isolation and attribute delineation, complemented by an LA module that mitigates the impact of these adaptations over successive generation steps. Through rigorous testing on T2I-CompBench and detailed visual case studies, our method demonstrates a substantial enhancement in handling complex multi-entity prompts, providing a promising avenue for future research in advanced image synthesis.

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
