# OpenReview forum: "EGGen: Image Generation with Multi-entity Prior Learning through Entity Guidance"
_acmmm.org/ACMMM/2024/Conference — MM2024 Poster_

### Official Review · Reviewer_wzmG · 2024-05-08

**Rating:** 4
**Confidence:** 3

**Summary:**

This article investigates multi-object generation, which is a significant research focus and also a current weakness in the field of visual generation. Stable diffusion is constrained by the encoder of CLIP, which can only extract the main part of the sentence, making it difficult to understand multiple objects. This article effectively addresses this issue.

**Strengths:**

The motivation is clear, the proposed method is effective, and it tackles a pain point problem.
The experiments are thorough.

**Limitations:**

If multi-object generation could be achieved solely at the level of stable diffusion without relying on large models, it might be a more meaningful endeavor.
The use of cross-attention editing methods has been extensively used in previous papers. While effective for the task at hand, its application in this context may lack inspiring significance for future research or substantial structural innovation.


In other words, while multi-object generation is a significant problem, addressing it at the level of CLIP or stable diffusion would be more impactful. However, employing a combination of large models and cross-attention may lead to some shortcomings in this work.
So I tend to give a Borderline Accept.

**Suitability:**

3

---

### Official Review · Reviewer_XVJW · 2024-05-09

**Rating:** 3
**Confidence:** 4

**Summary:**

This paper studies the the problem of inconsistent multi-entity representation (IMR) in text-to-image diffusion model, which is a valuable research challenge. The authors proposed to integrate the EGGen mechanism within cross-attention operations, which effectively improved
entity positioning and attribute accuracy while maintaining generation fidelity compared to the baseline method.

**Strengths:**

1. The studied problem is interesting and the authors proposed a relative effective method to alleviate the IMR problem.
2. Experimental results show less attribute leakage of the proposed methods compared to the other methods.
3. The paper is well organized and easy to read.

**Limitations:**

1. The generated images are not very realistic in fig.1, fig.7, etc. I think the main reason lies in the generating capabilities of the base model SD. It would be better to verify the proposed method on a more advanced model, e.g. sdxl.
2. In fig.7, attribute leakage still exists in the generated images, see the red and blue wall in the background of the second case.
3. This paper introduces HED soft edge as the prior guidance. Other method like Controlnet uses canny or HED edges to provide accurate location relationships. What's the difference between the proposed $L_{hed}$ with controlnet?

**Suitability:**

3

---

### Official Review · Reviewer_yAUZ · 2024-05-23

**Rating:** 4
**Confidence:** 3

**Summary:**

This paper first analyzes the problem of entity coupling and entity prematurity of existing text-to-image, which the author classifies as multi-entity representation (IMR) challenge. To solve these problems, the author proposes entity guidance generation mechanism, which is a divide-and-conquer mechanism, uses LLM to split text prompts and regions for input. In addition, three modules are proposed in this paper, such as ECA, GEA, and LA to achieve precise positional control and attribute accuracy in the generation of multiple entities.

**Strengths:**

1、This is a well-written paper, the questions raised are interesting and the technical points are easy to understand.

2、In this paper, a large number of experiments and analyses have been carried out to prove that the existing text-to-image generation method (SDv1.5) has entity coupling and entity prematurity problems, and the source of the problems is cross-attention.

3、This paper proposes a diffusion model of text-generated images for multi-entity generation. Its core idea is to decompose text prompt and regions to ensure the location and generation of a single entity.

4、This paper innovates the network structure，where the ECA for safeguarding each entity's uniqueness and correctness of the type, the GEA for guiding accurate entity positioning and attribute delineation, and the LA for preventing oversaturation and ensuring generation fidelity.

**Limitations:**

1、Although this article analyzes and summarizes the existing issues of entity coupling and entity prematurity, its technological innovation is slightly weaker.
The proposed model complements GLIGEN(CVPR2023) by adding two main modules ECA and GEA, as well as a loss function hed loss. However, the two modules are similar in implementation: Formula 5 and formula 6 follow the same computation format, the difference is that the former uses attention to compute features, the latter uses convolution to compute features, and then their results are dot-multiplied with masks.

2、The method of controlling the position of multiple objects in an image using decomposed text prompt and segmented masks has been used in DenseDiffusion (ICCV2023). This is similar to a divide-and-conquer mechanism proposed by the authors, who can include this reference in the paper.

3、The author uses a large number of visual experiments to analyze the problems of the existing methods, but does not demonstrate the effectiveness of the proposed module through visual experiments.

4、From the results of quantitative comparison, the proposed method has poor shape effect. However, the author used edge image as shape supervision for training, and did not conduct a separate ablation experiment to verify hed loss. The authors need to supplement the ablation experiments of hed loss to verify the role of this loss.

5、In Figure 9, I think the text prompt is the same as in Figure 1, but I find that there are a lot of messy flowers in the background of ECA+LA and ECA+GEA, which should be "A green garden". Why is this phenomenon? Are there any other examples of ablation experiments?

**Suitability:**

2

---

### Official Review · Reviewer_mmnM · 2024-05-24

**Rating:** 4
**Confidence:** 2

**Summary:**

The paper proposes to address the inconsistent multi-entity representation (IMR) challenge. Specifcially, it first reveals that the IMR challenges largely stem from the process of cross-attention mechanisms and introduces the entity guidance generation mechanism by integrating plug-in networks. The mechanism segments comprehensive prompts into distinct entity-specific prompts with bounding boxes, enabling a transition from multi-entity to single-entity generation cross-attention layers. It further introduces entity-centric cross-attention layers that focus on individual entities to preserve their uniqueness and accuracy, alongside global entity alignment layers that refine cross-attention maps using multi-entity priors for precise positioning and attribute accuracy. Additionally, a linear attenuation module is integrated to progressively reduce the influence of these layers during inference, preventing oversaturation and preserving generation fidelity. Extensive empirical results on the benchmark datasets demonstrate the effectiveness of the proposed method.

**Strengths:**

+ The paper proposes to decompose the multi-entity generation problem into single-entity generation sub-problems, which is interesting.
+ Extensive experiments demonstrate the effectiveness of this approach.
+ The ablative studies are thorough.
+ The presentation is clear and easy to follow.

**Limitations:**

Not necessarily limitations, mostly open-ended questions.
- It seems that the number of objects in the generated images (especially in the qualitative results) are mostly 2. How well does the proposed method work on cases with more than 2 objects?
- How well does the proposed method work on negation of a given concept, e.g., not red?
- I would like to hear about other reviewers' opinions regarding the limitations.

**Suitability:**

3

---

### Meta-Review · Area_Chair_cvKg · 2024-07-06

**Recommendation:** Accept (Poster)
**Confidence:** 4

**Metareview:**

This paper addresses the issue of inconsistent multi-entity representation (IMR) in the generated images of text-to-image diffusion models. All reviewers agreed that the problem is interesting, the proposed method to mitigate the IMR problem is effective, and the paper is well-written and easy to follow. While the reviewers noted that the technical novelty of the proposed method is not very high and there is room for improvement in its effectiveness, they concur that the paper meets the acceptance threshold.

It is strongly recommended that the authors incorporate the reviewers' feedback and rebuttal discussion into the final version of the paper to further enhance its comprehensiveness and quality.